# Independent Heating Performances in the Sub-Zero Environment of MWCNT/PDMS Composite with Low Electron-Tunneling Energy

**DOI:** 10.3390/polym15051171

**Published:** 2023-02-25

**Authors:** Yun Kyung Min, Taesik Eom, Heonyoung Kim, Donghoon Kang, Sang-Eui Lee

**Affiliations:** 1Department of Mechanical Engineering, Inha University, 100 Inha-ro, Michuhol-gu, Incheon 22212, Republic of Korea; 2KIURI Center for Hydrogen Based Next Generation Mechanical System, Inha University, 36 Gaetbeol-ro, Yeonsu-gu, Incheon 21999, Republic of Korea; 3Research Institute, PILETA Co., Ltd., Daejeon 34000, Republic of Korea; 4Korea Railroad Research Institute, 176 Cheoldo Bangmulgwan-ro, Uiwang 16105, Republic of Korea

**Keywords:** multi-wall carbon nanotube (MWCNT), joule heating, temperature coefficient of resistance, composite heater, activation energy

## Abstract

The structural stability of various structures (railroads, bridges, buildings, etc.) is lowered due to freezing because of the decreasing outside temperature in winter. To prevent damage from freezing, a technology for de-icing has been developed using an electric-heating composite. For this purpose, a highly electrically conductive composite film with multi-wall carbon nanotubes (MWCNTs) uniformly dispersed in a polydimethylsiloxane (PDMS) matrix through a three-roll process was fabricated by shearing the MWCNT/PDMS paste, through a two-roll process. The electrical conductivity and the activation energy of the composite were 326.5 S/m and 8.0 meV at 5.82 Vol% of MWCNTs, respectively. The dependence of the electric-heating performance (heating rate and temperature change) on the applied voltage and environmental temperature (from −20 °C to 20 °C) was evaluated. The heating rate and effective-heat-transfer characteristics were observed to decrease as the applied voltage increased, while they showed the opposite tendency when the environmental temperature was at sub-zero temperatures. Nevertheless, the overall heating performance (heating rate and temperature change) was maintained with little significant difference in the considered external-temperature range. The unique heating behaviors can result from the low activation energy and the negative-temperature (*T*) coefficient of resistance (*R*) (NTCR, d*R*/d*T* < 0) of the MWCNT/PDMS composite.

## 1. Introduction

The icing on buildings, structures (roads, railways, and bridges), and various conveyances (cars, trains, and aircraft) is the cause of various accidents. In the case of aircraft, aircraft icing occurs when supercooled water droplets freeze when the aircraft flies in clouds at temperatures below freezing point. Ice accretion on rotors, wing and tail surfaces of airplane increases flow resistance or causes sudden flow abnormalities, leading to devastating accidents of airplanes falling to the ground [1]. In winter, abnormal climate conditions such as sudden cold waves and heavy snow have recently occurred, and cause railroad accidents. Soil freezing or railroad contraction cause derailment and frostbite on the tracks that lift the ground, leading to serious accidents. Therefore, management of freezing in the facilities is needed for preventing serious accidents and reducing maintenance costs [2,3].

Electric-heating (joule-heating) materials and composites have been studied for the management and prevention of freezing [4]. Conventional electric-heating technology induces heating of structures themselves by implanting a conductive metal or alloy in the structures [5]. However, the repetition of heating and cooling inside such a structure has a disadvantage that weakens the durability of the structure. The technology of applying the electrically conductive composite film to the structure can overcomes the problems and thus has the advantage of maintenance. A conductive composite is composed of matrix material and electrically conductive fillers, and it is possible to appropriately control mechanical and electrical properties by controlling the selection and composition of the matrix and filler. Regarding this issue, recently studies were conducted to evaluate the heating performance by fabricating carbon-based polymer-composite film [6,7].

The conductive filler in the electrical-heating film has been studied, for example, metal, alloy, conductive metal oxides, carbon-based materials, conductive polymers, and hybrid conductive materials [8]. Carbon-based nanomaterials such as graphene and carbon nanotubes (CNT) are the most promising materials because they have high electrical-thermal properties, they are lightweight, and have excellent mechanical properties. CNT is a large molecule composed of a graphene sheet rolled like a cylinder. These CNTs have high thermal conductivity and excellent electrical conductivity, and composite materials are applied to various applications such as structures, sensors, and composites heaters [9,10,11]. CNTs can be divided into two types, depending on the structure of the wall; those composed of a single wall are called single-wall carbon nanotube (SWCNT), and those with two or more walls are called multi-wall carbon nanotube (MWCNT). SWCNTs have higher thermal conductivity, electrical conductivity, and better mechanical properties than MWCNTs, but MWCNTs have the advantage of economic efficiency, with easy mass production and low cost.

Electrically conductive composites using CNTs change their electrical properties depending on the temperature; when the electrical resistance decreases as the temperature rises it is called the negative temperature-coefficient of resistance (NTCR), and when the electrical resistance increases, it is called positive temperature-coefficient of resistance (PTCR) [12]. The activation energy of electron tunneling (*E_a_*) can be obtained through Arrhenius plotting for electrical resistance and temperature, which means the degree of tunneling or hopping of the obtained activation-energy charge carriers [13]. The activation energy of the conductive composite is governed by the electrical properties and concentration of the conductive filler and the matrix, and also the degree of dispersion and orientation of the filler. Low activation energy means that charge carriers are more easily transferred. There was a report for the activation energy for the bulk (6 meV), the individual (15 meV), and the bundles (0.234 eV) of MWCNTs [14], while polymer matrices have much higher electron-tunneling energy in epoxy (1.5 eV) and PDMS (polydimethylsiloxane) (0.73 eV) [6]. The much higher *E_a_* of the polymer compared to the fillers makes us try to enhance the degree of dispersion of the conductive filler, leading to direct contact or close distance between the fillers, in order to suppress the tunneling barrier. CNT/PDMS composites were reported to have activation energy in the range of 6 meV to 1.38 eV [14,15]. Decrease in temperature cause difficulty for the electron transfer in the composite materials, due to the nature of the NTCR character, and thus the independence of the electrical property regarding temperature needs to be secured by minimizing the electron-tunneling activation energy.

In this study, MWCNT/PDMS composite films were fabricated as electric-heating films attached on top of concrete brick, a representative element of infrastructures. For the dispersion of MWCNTs with a large aspect ratio (>10^4^) in the PDMS matrix, three-roll milling using mechanical shearing was employed [16]. The three-roll method, which can apply strong shear force without heating, can be the most suitable method for good dispersion of the highly viscous PDMS matrix with high MWCNT concentration. In film fabrication, a two-roll method was applied, because of its high viscosity, in order to produce the MWCNT/PDMS film under stable dispersion of MWCNTs. In order to evaluate the heating performance in environmental conditions mimicking cold weather in winter, the environmental temperature was set up in the range of −20 °C to 20 °C. The heating rate and temperature change were obtained through the time-vs.-temperature graph, and the heating characteristic growth-time constant and effective heat transferred under each condition were calculated. In addition, the activation energy for the electron tunneling was obtained through an Arrhenius plot of the temperature range. Based on the results, the fabricated composite was confirmed to be an ideal heating film in the sub-zero-Celcius-degree environment.

## 2. Materials and Methods

### 2.1. Materials and Fabrication of MWCNT/PDMS Composite Film

PDMS (Dow Corning, Sylgard 184, Midland, MI, USA) was used as matrix material, and MWCNTs (JEIO, Incheon, Republic of Korea), with a diameter of 6–9 nm and a length of 100–200 μm, were added to the matrix material at an amount of 10 wt% (5.82 vol%). The content of the MWCNTs was scanned in the range of 0.01, 0.05, 0.1, 1, and 10 wt% by considering the resistance level for the composite-film heater, and was determined in the processing window of the two-roll process used in this study by considering the viscosity level around 2000 Pa⋅s at the shear rate of 100/s.

The curing agent and PDMS prepolymer were mixed in a ratio of 1:10. The overall fabrication process of the MWCNT/PDMS composite is illustrated in Figure 1. In order to disperse MWCNTs in the PDMS matrix, the three-roll process was performed with a roll spacing of 10 μm for five passes and 5 μm for five passes, to disperse MWCNTs. Then, it was produced in the form of a film through a two-roll process, cured at 120 °C for 1 h, and then peeled off from the roll. The specimen size was cut to 65 mm in length and 160 mm in width, and the thickness was produced at a level of 200–300 μm. For electrodes, 50 μm thick copper tapes were attached onto the composite specimens, and silver paste was applied between the composite and the copper tape for stable contact; it was cured at 95 °C for 30 min and then 175 °C for 2 h, in order to prepare an electrode on the composite film for electrical-conductivity measurement and evaluation of heating performance.

### 2.2. Characterization and Performance Measurement

The microstructure of the composite was observed using a scanning electron microscope (SEM; S-4300SE, HITACHI, Tokyo, Japan) to evaluate the degree of dispersion of MWCNT in the PDMS matrix. Electrical resistance, (*R*), was measured using the four-point probe geometry (Mitsubishi Loresta-GP (MCP-T610) resistivity meter (Mitsubishi, Tokyo, Japan). The electric-heating performance was conducted by applying DC 5 V, 9 V, and 12 V at the environmental temperatures (*T_Env_*) of −20 °C, −10 °C, 0 °C, and 20 °C, when the MWCNT/PDMS composite was placed on a concrete brick. Temperature change over time was measured using a thermocouple (K-type) attached to the center point of the specimen surface, and measurement data were collected through DAQ (NI9213, National Instrument Co., Austin, TX, USA). In addition, to check the heat distribution over the entire specimen, thermal images of the surface of the specimen were obtained using an IR camera (TESTO 882, TESTO Co., Lenzkirch, Germany). To observe the change in resistance according to temperature, the change of resistance was measured in real-time by placing the MWCNT/PDMS film in a thermal chamber and by increasing the temperature at intervals of 10 °C from −20 to 60 °C. When the resistance value was determined, the temperature of the thermal chamber was maintained for approximately 30 min, so that the temperature of the sample was taken to be reached in a steady state.

## 3. Results and Discussion

### 3.1. Microstructures of MWCNT/PDMS Composite Film

Carbon nanomaterials such as CNT, graphene, and carbon black are easily agglomerated, due to their strong van der Waals interaction. Therefore, uniform dispersion in a solution or matrix is possible by exfoliating with strong external forces or creating repulsive forces between nanomaterials through surface modification. The three-roll milling technique is a mass-scalable tool for uniformly dispersing a high content of filler in the matrix by applying high shear force on agglomerated fillers. The gaps between rolls, the rotation speed of the rolls, and the number of process passes was monitored and optimized [17]. The number of observed MWCNTs clearly increased with its content (Figure 2a–e). The MWCNT was not easily detected below 0.1 wt%, but began to be clearly observed from 1 wt%. MWCNTs are observed to be debundled and uniformly dispersed in the PDMS matrix, as shown in Figure 2e. Figure 2f shows the as-received MWCNT.

### 3.2. Electrical Conductivity of MWCNT/PDMS Film

In order to analyze the electrical properties of the MWCNT/PDMS at room temperature and the uniformity of the specimen, the electrical conductivity was calculated through sheet-resistance measurement.

The electrical conductivity of the sample increased from 0.1 mS/m to 326 S/m as the MWCNT content increased from 0.01 to 10 wt% (Figure 3). When the content was increased from 0.05 to 0.1 wt%, the electrical conductivity increased rapidly, which means that the percolation threshold was formed in this range. To confirm the exact percolation threshold, classical percolation theory was applied as follows.
*σ* = *σ*_0_ (*Φ* − *Φ_c_*)*^t^*(1)
where *σ*_0_ is the factor depending on the intrinsic conductivity of the MWCNT, *Φ* is the content of the MWCNT, *Φ_c_* is the percolation threshold, and *t* is a critical exponent. As a result of plotting the electrical conductivity according to Equation (1), *Φ_c_* of the MWCNT/PDMS composite dispersed through our process was obtained as 0.029 wt%. In general, it was known that the percolation threshold of CNT/polymer composites with the aspect ratio (AR) of CNT near 1000 was approximately 0.1 wt% (~1/AR) when optimized-dispersion methods were applied [17]. Compared to the relatively high *Φ_c_* of other previously reported MWCNT/PDMS composites [6,18], since the aspect ratio of CNT used in this study is as high as over 10,000, a percolation threshold was formed relatively lower than 0.1 wt%. In addition, the *t* value of 2.27 indicates that a quasi-3D network structure of MWCNT formed in the PDMS matrix. This means that the three-roll for dispersion and the two-roll for film formation are very suitable methods for the dispersion of the MWCNTs with a high aspect ratio in the PDMS matrix. As a result, a composite film was fabricated with a *σ*_0_ value of level 10^5^ S/m and a very low anisotropy ratio of electrical conductivity (axial/circumferential = 1.02).

The electrical conductivity of the composites had a value of 326.5 S/m at 10 wt% (5.82 Vol%), averaged at 330.0 S/m in the axial direction and 322.9 S/m in the circumferential direction of a roller in the two-roll film-fabrication setup, displaying little preferential orientation of the MWCNTs from shearing the uncured MWCNT pastes.

The MWCNT/PDMS composite-film formation through the two-roll process showed different features from other methods such as bar coating, doctor blade coating, and the use of a film applicator. In particular, when the viscosity is very high, such as the PDMS containing a high content of MWCNT, the two-roll method is a very suitable for film fabrication. Furthermore, the CNT composite film formed through the two-roll process has a relatively in-plane isotropic alignment of CNTs (Figure 4).

It was reported that the in-plane anisotropy ratio of electrical conductivity expressed as the ratio of the alignment direction and its vertical or transverse direction was 1.97 and 2.78 at a 5.66 wt% content of MWCNTs synthesized using the flame-synthesis (FS) process and spraying-and-thermal-decomposition (STD) process, respectively, when using a film applicator [18,19]. The result indicates that MWCNT alignment can be affected by the intrinsic properties of MWCNT, such as aspect ratio, or diameter and length, and degree of bundling, which is controlled by the synthesis method of the nanocarbon. Aside from the intrinsic geometric character of the MWCNTs, there is the difference in the fabrication process, which are the two-roll process in this study and the film applicator. In the two-roll process, there is shear flow in the circumferential direction, while there is an extensional flow in the roll-axis direction. The balance in the two flows, together with the geometry of the filler, can be attributed to the quasi-isotropic in-plane electrical conductivity.

The electrical conductivity of the electrically conductive composites depends on the intrinsic conductivity and interfacial character of the filler and matrix, shape and volume fraction of the filler, and the orientation or alignment of the filler. Other types of MWNT/PDMS composites showed 301 S/m at 5.7 Vol%, 43.7 S/m at 5.0 Vol%, and 0.04 S/m at 6.5 Vol%, and MWNT/VMQ (vinyl-methyl silicone rubber) showed 8 S/m at 5.83 Vol% in electrical conductivity [20,21,22,23]. Compared to the values in the above-mentioned studies, the electrical conductivity reached in this study is at a high level. Therefore, it can be confirmed that the material system and the processing in this study can be an effective solution for fabricating highly electrically conductive polymer composites.

### 3.3. Electric-Heating Performances and Characteristics

To investigate the electric-heating performance of the MWCNT/PDMS composite, voltages of 5 V, 9 V, and 12 V were applied at different environmental temperatures. The surface temperature of the MWCNT/PDMS film heater was measured by attaching a thermocouple onto the sample, and also visualized thermal images were obtained through a thermal-imaging camera (Figure 5a).

Figure 5b is an image taken with a thermal-imaging camera of the temperature change of the sample according to the applied voltage at room temperature (20 °C). The higher the voltage, the higher the sample-temperature-rise rate and the final reached temperature, which can be explained by Joule’s law. In addition, a uniform temperature change was observed throughout the sample, regardless of the applied voltage, as shown in Figure 5b. The increase in heating performance by applied power (*P*) can be understood through the relationship between Joule heating and heat capacity (*c∙m*, where *c* is specific heat and *m* is mass) in Equation (2) [24,25,26].
*P* = *V*^2^/*R* = *I*^2^ ∙ *R* = *c* ∙ *m* ∙ ∆*T*(2)
where ∆*T* is the temperature change in the composite brought about by the applied power, *V* and *I* are the applied voltage and current, and *R* is the electrical resistance of the composite. Through the relational Equation (2), the temperature change of the sample increases as the applied voltage increases. The temperature rise of the sample may represent a discrepancy from the theoretical value, which is contributed to by the combined effect of the surrounding environmental temperature, heat diffusion from the heating element into the surroundings, energy loss when converting power to heat, and the intrinsic heating characteristics of the composite with regard to temperature [27]. This observation is consistent with other compositions of MWCNT/cellulose [28].

Figure 6b–e are the graphs of the measured temperature change through the thermocouple placed on the MWCNT/PDMS surface, according to the applied voltage and the elapsed time. Since the electric-heating experiment was conducted when the composite temperature reached the environmental temperature, the initial temperature was taken as the environmental temperature. Although the initial temperature and the final temperature were different for the given *T_Env_*, the temperature increment was observed at a similar level.

From the graphs of the elapsed time vs. temperature, the heating rate was obtained from the slope in the initial-temperature-rise region, and Δ*T* was obtained from the difference between the initial temperature and the maximum temperature, as shown in Figure 7. The heating rate increased as the applied voltage increased, and the average heating rates were 3.62, 12.53, and 22.93 °C/min at 5 V, 9 V, and 12 V, respectively. In particular, when 5 V was applied, as the environmental temperature was lowered to 20 °C and −20 °C, the heating rate increased remarkably, from 2.7 to 4.3 °C/min, respectively. In the case of 9V and 12V, there was no significant difference in heating rate even when the environmental temperature was decreased. In contrast to the heating rate, in the case of an applied voltage of 5 V, the temperature changes showed 7.3, 8.6, 7.4, and 8.0 °C at 20, 0, −10, and −20 °C, respectively, indicating little effect of the environmental temperature. When 9 V and 10 V were applied, the temperature change slightly decreased as the environmental temperature decreased. The differences in temperature change between 20 °C and −20 °C of the environmental temperature were only 4.5 °C and 4.0 °C in the cases of 9 V and 12 V, respectively. Due to this heating rate and temperature-change tendency, the temperature of the composite rapidly increased within 5 min and then became saturated. When the environmental temperature decreased, the temperature rise decreased, but the temperature rose more than 45 °C at 12 V, and therefore has sufficient heating performance to be applied as de-icing material.

To investigate the heating characteristics of the MWCNT/PDMS composite film, the electric-heating characteristics were analyzed by defining three temperature ranges over time: a ramp-up region, a steady-state temperature region, and a cooling one [26,29]. In the results of this experiment, the heating region was set to 0–5 min, and the steady-state temperature region was set to 5–20 min (Figure 6b). The heating-characteristic growth-time constant (*τ_g_*) during temperature increase due to the power applied in the ramp-up region can be expressed as follows.
(*T_t_* − *T_i_*)/(*T_m_* − *T_i_*) = 1 − exp(−*t*/*τ_g_*)(3)
where *T_t_* are the temperature at time *t*, *T_i_* are the initial temperature, and *T_m_* are the maximum temperature. *τ_g_* is an indication of response time for initial heating.

The thermal stability of the MWCNT/PDMS composite material can be confirmed through the amount of heat transfer in the steady-state temperature region. The effective heat transferred by convection and radiation-per-unit temperature, *h_r+c_*, is expressed as the steady-state current (*I_c_*) and the initial applied voltage (*V_i_*), as follows.
*h_r_*_+*c*_ = (*I_c_V_i_*)/(*T_m_* − *T_i_*)(4)

Figure 8 is the graph showing the relationship between average *τ_g_* and *h_r+c_*, according to the applied voltage and environmental temperature. Table 1 shows *τ_g_* and *h_r+c_* values under each condition in more detail. From Figure 8a, *τ_g_* decreased from 1.58 to 1.23 min as the applied voltage increased. As the applied voltage increased, the power increased, the heat generated increased, and thus *T_m_* and *T_t_* increased. As a result, the (*T_t_* − *T_i_*)/(*T_m_* − *T_i_*) value increased, leading to an improvement in the response time for the initial heating region. In more detail, *τ_g_* tended to decrease as the applied voltage increased at a temperature over 0 °C, but showed a reverse *τ_g_*-value tendency at under 0 °C. In particular, *τ_g_* was 2.2 min at *T_Env_* = 20 °C at 5 V, but the value decreased to 1.0 min at *T_Env_* = −20 °C. In the case of −20 °C, the same level of the *τ_g_* values were observed, regardless of the applied voltage. This tendency was similar to the heating rate described above. In other words, the initial temperature rises more rapidly at low temperatures, which is advantageous for application to the actual de-icing field. The values of effective heat transfer through convection and radiation ranged from 2.9 W/°C to 3.1 W/°C, regardless of the environmental temperature.

### 3.4. Resistance-and-Temperature Relationship of MWCNT/PDMS Film

The resistance-vs.-temperature characteristics of conductors can be divided into two types: NTCR (negative temperature-coefficient of resistance), which shows resistance decrement as the temperature rises, and PTCR (positive temperature-coefficient of resistance), which shows resistance increment as the temperature rises. In order to evaluate thermal-resistance characteristics, the MWCNT/PDMS composite film was placed in a thermal chamber, to simulate the ambient temperature from −20 °C to 60 °C, and then the resistance value was measured in real time by waiting for 30 min until the temperature of the sample was stabilized (Figure 9a). 

The TCR of a composite generally depends on the intrinsic properties of fillers and the matrix, and the shape and volume fraction of fillers [19,20,21,22]. The MWCNT/PDMS composite showed NTCR characteristics (Figure 9b). The measurement was conducted successfully, as there was no significant time difference between the temperature change and the resistance change. When the resistance and temperature were plotted as the average of the resistance values in each constant-temperature setup region, a linear relationship was obtained, as shown in Figure 9c. Electrical characteristics can be obtained through the Arrhenius equation, as follows.
(5)$$R=R_{o}^{\prime }\exp\left(\frac{E_{a}}{k_{B}T}\right)\to \ln\left(R\right)=\frac{E_{a}}{k_{B}T}+\ln\left(R_{o}^{\prime }\right)\;\;\;\;\;\;\;\;\;\;\;\;\;\left(\sigma \propto \frac{1}{R}\right)$$
where *R^’^*_0_ is the constant pre-exponential factor, *E_a_* is the activation energy for electron tunneling of composites, *k_B_* is Boltzmann’s constant, and *σ* is the electrical conductivity. *E_a_* can be calculated from the slope of the graph. The Arrhenius plot of the MWCNT/PDMS composite has high linearity. The *E_a_* value of the MWCNT/PDMS composite was 8.0 meV, which is significantly lower than that of the previously reported CNT composites by several tens of meV. The *E_a_* values in the literature were 0.032 eV for SWCNT/glass (2.54 × 10^−4^ mg%) composite [30], 0.019 eV for MWCNT/PDMS (7.0 Vol%) composite [31], and 0.022–0.062 eV for PEDOT:PSS/SWCNT (12 and 16 wt%) composite [32]. In particular, the aspect ratio of MWCNT in this study (dia.: 7–9 nm, length: 100–200 µm) was higher than MWCNT/PDMS dia.: 10–15 nm, length: 1–10 µm) [31]. The activation energy of MWCNT has been reported as 0.234, 0.015, and 0.006 eV, depending on the dispersion state of the individual, bundle, and bulk MWCNTs [14,16]. Depending on the dispersion state of the MWCNTs, differences in tunneling have occurred between tube to tube and shell to shell. The low *E_a_* values of MWCNT/PDMS composites are intermediate, between the values of the individual CNT and the bulk CNTs. Although dispersion of high aspect-ratio nanomaterials is relatively more difficult, the high electrical conductivity and the low *E_a_* values indicate that the MWCNTs were well and individually dispersed in the PDMS, and formed the closely networked arrangement of the conductive fillers.

The composite material with NTCR properties has the characteristic that the resistance increases as the temperature decreases and the heating efficiency decreases, according to Equation (2). The MWCNT/PDMS composite in this paper also showed NTCR characteristics, −1.25 × 10^−2^ Ω/K. MWCNT/PDMS (7.0 Vol%) was −2.07 × 10^−2^ Ω/K, even in the higher filler content [32]. In addition, because it has a low *E_a_* value, there is no significant decrease in resistance in the temperature range of the actual operating environment (Figure 9d). Therefore, since there was little significant difference in electrical characteristics in the range of *T_Env_* = −20 to 20 °C, the electric-heating characteristics of the MWCNT/PDMS composite were at a similar level, regardless of the environmental temperature. Figure 9d shows the theoretical voltage and corresponding output for our MWCNT/PDMS composite and the MWCNT composite of other papers with higher *E_a_* (0.019 eV) [32]. When the temperature was lowered from room temperature (20 °C) to sub-zero conditions (−20 °C), our MWCNT/PDMS composite showed a power decrease of approximately 5%, whereas another MWCNT composite with the three-roll milling process and the bar coating had a power decrement of approximately 11%. The difference in the power change may result from the film-forming process (the two-roll process vs. hot pressing [32]), the filler loading (5.82 Vol% vs. 7.0 Vol% [32]), or the intrinsic geometrical character of the MWCNTs (6–9 nm vs. 10–15 nm [32] in diameter, and 100–200 μm vs. 1–10 μm in length, respectively. In addition, the heat capacity of the SWCNT composites generally tended to decrease with the temperature decrease [33,34]. In other words, even if the resistance value slightly increases in the low-temperature range, the temperature increase could be maintained at a similar level, due to the lowered heat capacity, according to Equation (2). For those reasons, although the environmental temperature was low, the temperature change (∆*T*∝*P*) and *h_r+c_* (*h_r+c_*∝1/Δ*T*) were scarcely different from the temperature change at room temperature, as observed in Figure 7. This thermal-resistance characteristic can be a great advantage, because the regular output can be maintained in more controllable resistance and power range, regardless of the external temperature in real-field applications.

## 4. Conclusions

In this study, MWCNTs were successfully dispersed in the PDMS matrix through the three-roll process. By using this mixture, the composite heating film for de-icing was fabricated using the two-roll process. The electrical conductivity of the MWCNT/PDMS was 307 S/m at a 5.82 vol% of CNT content at room temperature. The applicability of MWCNT/PDMS composite films for de-icing application was investigated by measuring electric-heating performances from room temperature (20 °C) to cold environmental temperature (−20 °C). In the case of the electric-heating characteristics according to the environmental temperature, as the environmental temperature decreased, the maximum temperature (*T_m_*) and the initial-heating-reaction time (*τ_g_*) were decreased, whereas the effective heat transferred (*h_r+c_*) was not significantly changed. The *τ_g_*, according to the applied voltage, decreased when the voltage was increased at environmental temperatures above 0 °C, but showed the opposite trend at sub-zero temperatures; *h_r+c_* also showed a similar trend, but the differences were not significant enough. The *E_a_* value obtained through the Arrhenius plot was 8 meV, which was very low and means that MWCNTs were successfully dispersed in the PDMS matrix. In addition, this kind of low electron-tunneling barrier led to the result that the characteristics of electric heating were not significantly affected by environmental temperature. Therefore, it can be concluded that the MWCNT/PDMS composite film has potential as a de-icing element for aerial or civil structures in a cold environment. In addition, the application of the materials can be extended to heaters of a cylindrical shape [15,19] with FGM (functionally-graded materials) [35,36], piezoresistive [11,37] or temperature [38] sensors, thermoelectric generators [39], and EMI (electromagnetic-interference) shielding [28,40,41], by using high electrical conductivity, a low percolation threshold, or electron-tunneling energy. Together with the effect of environmental temperatures on de-icing and the composite behaviors conducted in this study, further studies on ice nucleation and growth [42,43] on the surface of de-icing composites can enrich the de-icing research field.

## Figures and Tables

**Figure 1 polymers-15-01171-f001:**
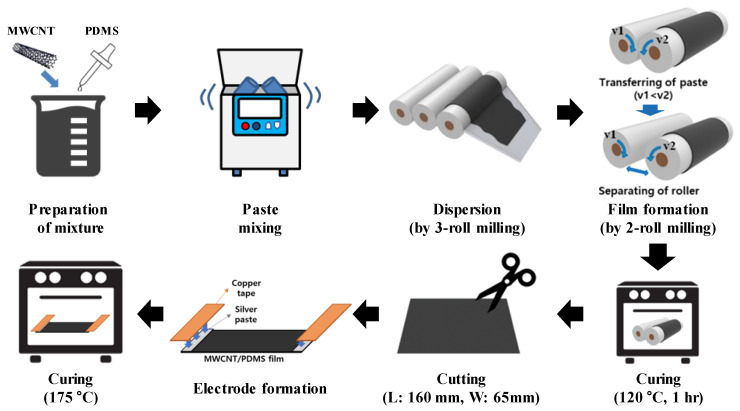
A schematic of fabricating MWCNT/PDMS composites.

**Figure 2 polymers-15-01171-f002:**
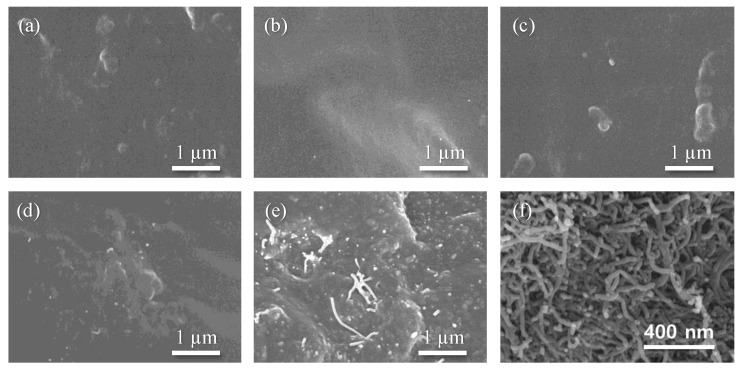
SEM images of MWCNT/PDMS composites of (**a**) 0.01, (**b**) 0.05, (**c**) 0.1, (**d**) 1, and (**e**) 10 wt% MWCNT. (**f**) SEM image of as-received MWCNT.

**Figure 3 polymers-15-01171-f003:**
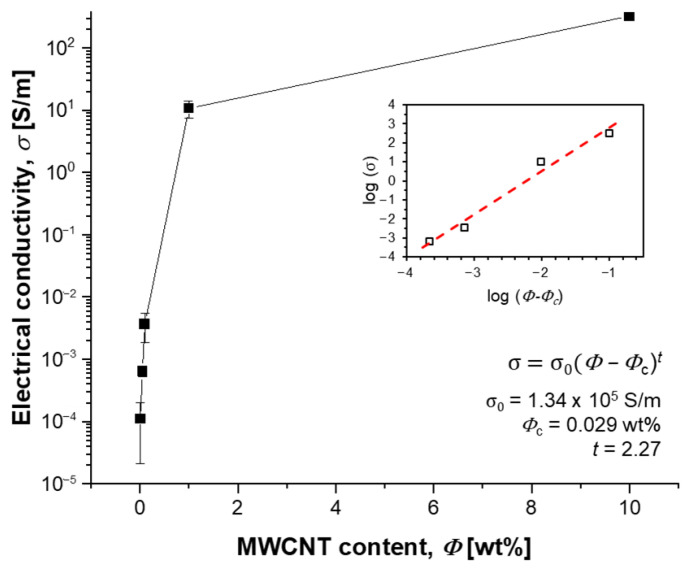
Electrical conductivity of MWCNT/PDMS composites in relation to MWCNT content. (Inset: Linear fit for determining the theoretical values of *Φ_c_* and *t*.).

**Figure 4 polymers-15-01171-f004:**
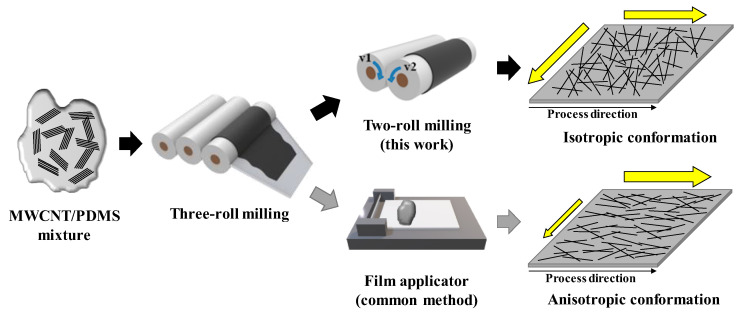
Schematic of MWCNT distribution conformation depending on the film-fabrication process.

**Figure 5 polymers-15-01171-f005:**
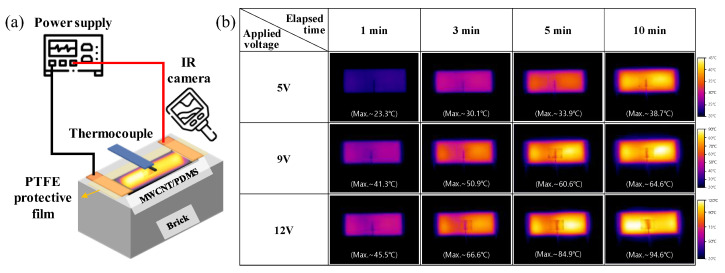
(**a**) Schematic measurement setup of electric-heating performance for MWCNT/PDMS on brick. (**b**) IR-camera image of MWCNT/PDMS film, according to time and applied voltage at *T_Env_* = 20 °C.

**Figure 6 polymers-15-01171-f006:**
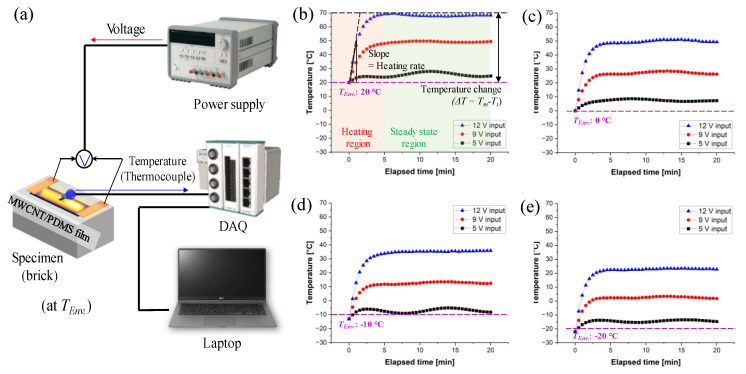
(**a**) Electric-heating setup and performances of MWCNT/PDMS composite depending on applied voltage (5, 9, and 12 V) and in (**b**) 20, (**c**) 0, (**d**) −10, and (**e**) −20 °C of environmental temperature (*T_Env_*).

**Figure 7 polymers-15-01171-f007:**
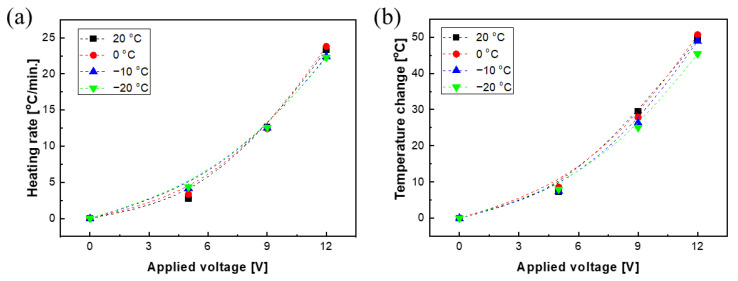
Heating characteristic of MWCNT/PDMS under different environmental temperatures; (**a**) heating rate and (**b**) temperature change.

**Figure 8 polymers-15-01171-f008:**
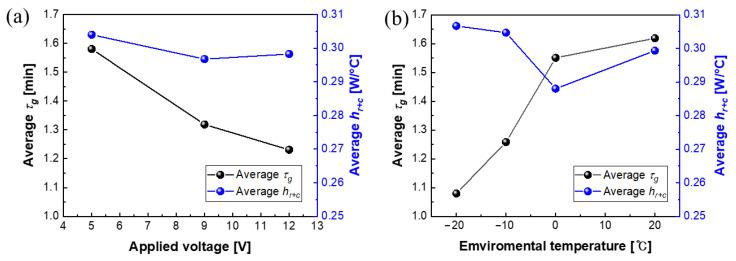
Average change in the heating-characteristic growth-time constant (*τ_g_*) and the effective heat transferred (*h_r+c_*) with regard to (**a**) applied voltage and (**b**) environmental temperature.

**Figure 9 polymers-15-01171-f009:**
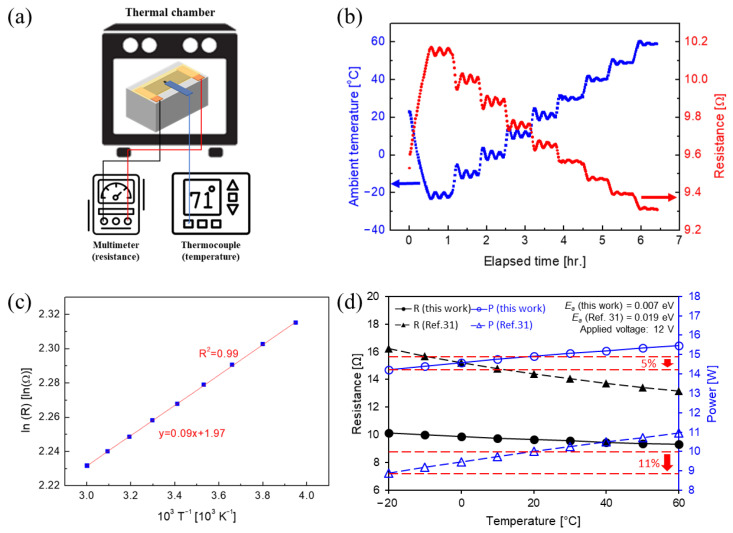
(**a**) Schematic of resistance-measurement setup for MWCNT/PDMS film on brick, depending on ambient temperature. (**b**) Changes in ambient temperature and resistance of MWCNT/PDMS film, according to the elapsed time. (**c**) Arrhenius plot of MWCNT/PDMS film, depending on ambient temperature. (**d**) Changes in theoretical resistance and power, according to temperature [31]. Reprinted/adapted with permission from Ref. [31]. Copyright 2016, copyright AIP Publishing.

**Table 1 polymers-15-01171-t001:** Characteristic parameters (*τ_g_*: the heating-characteristic growth-time constant, *h_r+c_*: the effective heat transferred) for electric-heating performance of MWCNT/PDMS composite film under different applied loads.

Environmental Temperature [°C]	Applied Voltage [V]	*τ_g_* [min]	*h_r+c_* [W/°C]
20	5	2.162	0.319
	9	1.490	0.280
	12	1.204	0.299
0	5	2.002	0.292
	9	1.386	0.287
	12	1.264	0.285
−10	5	1.142	0.315
	9	1.287	0.304
	12	1.345	0.295
−20	5	1.015	0.290
	9	1.112	0.316
	12	1.111	0.314

## Data Availability

Data sharing not applicable.

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
