# Peer review of "Independent Heating Performances in the Sub-Zero Environment of MWCNT/PDMS Composite with Low Electron-Tunneling Energy"

_polymers, 2023, doi:10.3390/polym15051171_

Round 1

Reviewer 1 Report (Previous Reviewer 1)

As the authors have addressed all the reviewers' comments, I am pleased to recommend it to be published in present version.

Author Response

Thank you for your positive review.

Reviewer 2 Report (New Reviewer)

Manuscript ID: polymers-2226928

Title: Independent Heating Performances in the Sub-zero Environment of MWCNT/PDMS Composite with Low Electron Tunneling Energy

The authors investigate MWCNTs dispersion in the PDMS matrix through the three-roll process. By using this mixture, the heating composite film for deicing was fabricated by the two-roll process. The subject addressed in this article is worthy of investigation and the information presented is new. Also, the conclusions are supported by the data, and the organization of the manuscript is appropriate. After addressing the following comments, the present work can be accepted for publication:

  ·         According to the experimental tests, has the uncertainty analysis been done?

·         Has the repeatability of the experiments been checked?

·         It is necessary to explain the characteristics of the test equipment in more detail.

·         A validation benchmark must be presented to ensure the accuracy of the results.

Functionally graded materials are the new generation of composite materials which overcome the issue of thermal resistance and also delamination in multilayer structures. It is suggested to introduce the thermal FGM materials for readers' information. Referring to the following papers will be useful: 10.1016/j.icheatmasstransfer.2019.104280; 10.1007/s10973-020-09482-5

Author Response

Thank you for your review.
We positively received the comments and wrote the response to this in the attachment.

Reviewer 3 Report (New Reviewer)

The authors have presented a MWCNT/PDMS composite fabricated through two-roll process for de-icing surfaces in winter to maintain structural integrity and stable operation. Electrical properties of the composite materials include threshold loading required for percolation and heating performance in the temperature range of – 20 oC to 20 oC have been investigated. The authors report that the materials show opposite heat transfer characteristics at sub-zero temperatures compared to above 0 oC. Additionally, the heating rate and temperature tests show minimal effects on the heating performance. Overall, the article is well written with detailed description of improved properties in the proposed materials. Further clarification of various points and new experiments below will strengthen the paper to recommend for publication in Polymers.  

Major Comments (Major scientific and technical concerns)

1. Schematics of test setups should be included for additional clarity and the actual photographic images of materials and test setups should be presented whenever possible. Please include the images of the material itself and while it is being tested either in the main figures or in a supplementary file.

2. The current study has good characterization experiments but lacks in illustrating a successful application of the materials. It would be helpful for the readers to put the materials in the context of a de-icing scenario and showcasing the actual performance to make this paper more comprehensive and impactful. For example:
a. https://www.sciencedirect.com/science/article/pii/S000862232100628X

b. https://www.sciencedirect.com/science/article/pii/S0266353819312576

c. https://www.sciencedirect.com/science/article/pii/S0266353819325874

d. https://www.sciencedirect.com/science/article/pii/S0008622318303981

3. In line 193, … has a relatively “anisotropic” alignment? Should this be anisotropic or isotropic?

4. What is the stiffness and stretchability of the composite? Can it be attached to different geometries or just flat surfaces?

5. All the figure components in each figure should be referenced in the main text, not just the whole figure. For example, for Figure 6, only component a was referenced. Please check this and correct for all figures.

6. In Figure 2, there are six images but only five were labeled in the caption. Correct this and change the description in main text appropriately.

Minor Comments (Minor concerns)

1. Minor grammatical errors should be corrected.

2. In line 69, what is economic efficiency? Is it cheaper to fabricate or cheaper to incorporate in polymers? Please consider revising this.

3. In line 154, ….as MWCNT content “was increased”? Please correct.

4. Line 285, Tt was written twice.

5. Please change deicing to de-icing throughout the text.

6. Correct second temperature on line 125.

Author Response

Thank you for your detailed comment.

We have accepted your request to the possible, and please check the attached document for details.

Round 2

Reviewer 2 Report (New Reviewer)

All the comments are addressed.

This manuscript is a resubmission of an earlier submission. The following is a list of the peer review reports and author responses from that submission.

Round 1

Reviewer 1 Report

The authors described a MWCNT/PDMS composite with low electron tunneling energy independence on environmental temperature for heating. However, the content and quality of manuscript can not meet the standard of polymers. So I suggest it to be rejected, and offer the following comments.

1. The method for fabrication the MWCNT/PDMS composite is regular and commen. So, in the section of introduction, the innovation of the article should be further clarified.

2. Lack of ratio optimization for MWCNT/PDMS.

3. The machenism of the independence on environmental temperature should be discussed in detail.

4. Some proper cases for applications may be added.

5. Moderate English changes are required.

Reviewer 2 Report

The authors have addressed all my previous questions and this manuscript can be accepted now.

Reviewer 3 Report

Authors fails to address to the key concern of the reviewers. A more careful review found the activation energy authors obtained is wrong. Therefore, it cannot be published.

1, Reviewers show concern about the optimized loading of CNTs. However, this was not addressed in the revised version. As authors point out, higher conductivity leads to better heating performance and the CNT loading can be adjusted from 5 to 15 wt.%. However, they just report one data point.

2, The calculation of activation energy is wrong. Fig 8b in revised version, the resistance changes over 10 % from temperature range of 80oC, however, the activation energy is only 0.019eV. Authors directly obtained the value based on Fig 8d where unit is 1000 K-1. This reduced the true value by 1000 times. The actual value is 1.9eV! The fitting should be against the T-1 while 1000T-1.